# Magnetic Nanocomposites of Coated Ferrites/MOF as Pesticide Adsorbents

**DOI:** 10.3390/molecules28010039

**Published:** 2022-12-21

**Authors:** Savvina Lazarou, Orestis Antonoglou, Stefanos Mourdikoudis, Marco Serra, Zdeněk Sofer, Catherine Dendrinou-Samara

**Affiliations:** 1Laboratory of Inorganic Chemistry, Department of Chemistry, Aristotle University of Thessaloniki, 54124 Thessaloniki, Greece; 2Department of Inorganic Chemistry, University of Chemistry and Technology Prague, Technická 5, 16628 Prague, Czech Republic

**Keywords:** nanoadsorbents, magnetic nanomaterials, magnetic metal–organic frameworks, UiO-66, water remediation, pesticides

## Abstract

Magnetic metal–organic frameworks (MMOFs) are gaining increased attention as emerging adsorbents/water remediation agents. Herein, a facile development of novel MMOFs comprised of coated ferrite nanoparticles (MNPs) and UiO-66 metal–organic framework is reported. In specific, coated Co- and Zn-doped ferrite magnetic nanoparticles were synthesized as building block while the metal–organic framework was grown in the presence of MNPs via a semi-self-assembly approach. The utilization of coated MNPs facilitated the conjugation and stands as a novel strategy for fabricating MMOFs with increased stability and an explicit structure. MMOFs were isolated with 13–25 nm crystallites sizes, 244–332 m^2^/g specific surface area (SSA) and 22–42 emu/g saturation magnetization values. Establishing the UiO-66 framework via the reported semi-self-assembly resulted in roughly 70% reduction in both magnetic properties and SSA, compared with the initial MNPs building blocks and UiO-66 framework, respectively. Nonetheless, the remaining 30% of the magnetization and SSA was adequate for successful and sufficient adsorption of two different pesticides, 2,4-Dichlorophenoxyacetic acid (2,4-D) and 2,4,5-Trichlorophenoxyacetic acid (2,4,5-T), while the recovery with a commercial magnet and reuse were also found to be effective. Adsorption and kinetic studies for all three MMOFs and both pesticides were performed, and data were fitted to Langmuir–Freundlich isotherm models.

## 1. Introduction

Moving on from the illustrious perspectives of magnetic nanoparticles (MNPs) and their plethora of applications, research and development has now shifted to more advanced magnetic nano-architectures (MNAs) [1,2,3,4]. These innovative materials retain and can even augment the magnetic properties of their building blocks while displaying additional characteristics such as delivery of molecular cargos, adsorption capacity, desired interaction with biomolecules, and preferred rheological/in-suspension behavior. All these added traits are governed by the specific design of the MNA that can lead potentially to an abundance of shapes, sizes, structures, and surface chemistries. As a result, these new-age materials exhibit multimodal action, greater overall potential and expanded applicability.

Amongst various applications, MNAs are currently investigated as particularly promising water remediation agents [5,6,7,8,9,10] with the obvious benefit being the ease of recovery and reuse via external magnets. While this advantage is provided by the MNP building blocks, the adsorption properties of the MNPs may be insufficient or even totally absent. On the other hand, conventional adsorbents, such as zeolites [11,12,13] and/or novel engineered materials such as metal–organic frameworks (MOFs) [14,15,16,17,18], have great porosity and tunable pore sizes, endowing highly selective adsorption capacity of hazardous contaminants from the environment. However, these adsorbents are difficult to be recycled from the mixture solution. To overcome these problems, the combination of MNPs with MOFs, under the tag of magnetic metal–organic frameworks (MMOFs) is gaining increasing attention for hazardous contaminant removal from the environment [19,20,21,22].

Various fabrication routes and strategies have been reported and diverse MMOFs have been isolated by each pathway [19,20,21,22]. However, in many of these cases, over 90% diminish in either the magnetic properties (provided by the MNPs) or the specific surface area (SSA, endowed by the MOF) cannot be avoided due to the altered ordering of the newly developed structure. It is also worth mentioning that stability of the MMOF in aqueous solutions can also stand as a challenging issue as dissociation and/or slow release of their metal ions and organic linkers may occur. Moreover, the synthetic economic cost of the MMOF can be prohibitive for transfer to the industrial scale if complex, multifaceted and multistep procedures are employed for the design of either the building blocks (MNPs and/or MOF) and/or the final MMOF nano-architecture.

Herein, in continuation of our endeavors on the synthesis of MNPs [23,24,25,26,27] and design of MNAs [28,29,30,31], we report on the fabrication of three different MMOFs as pesticide removal agents in water remediation, using coated Co- and Zn-doped ferrite nanoparticles as the MNP building blocks and UiO-66 as the MOF framework. Amine and/or PEG-ylated coated MNPs were synthesized following protocols from our previously reported routes utilizing green, facile, one-step syntheses in autoclaves. In that manner, coated MNPs with different sizes, surface characteristics and magnetization values were isolated. The construction of the MMOFs was achieved by a semi-self-assembly development of UiO-66 in the presence of MNPs. Keeping the same MOF in all cases, the influence of the primary MNP building blocks (different sizes, physicochemical properties and surface characteristics)was investigated. UiO-66 was also synthesized in the absence of MNPs for comparative purposes. MMOFs were characterized by means of scanning electron microscopy (SEM-EDS), X-ray diffraction (XRD), infrared spectroscopy (FTIR) as well as thermogravimetric analysis (TGA) to determine the composition, morphology, crystallite size, structure and organic content. Magnetic properties were evaluated with a vibrating sample magnetometer (VSM) while the specific surface area (SSAs) was measured via N_2_-porosimetry/BET theory. Adsorption efficiency was studied for two different pesticides, 2,4-dichlorophenoxyacetic acid (2,4-D) and 2,4,5-trichlorophenoxyacetic acid (2,4,5-T), with the aid of magnetic separation, whilst the quantification of the adsorption capacity was achieved via ultra-violet visible spectroscopy (UV-Vis). Results were fitted to both the Langmuir and Freundlich isotherm models. Moreover, pseudo first order (PFO) and pseudo second order (PSO) kinetic studies were also employed to study the adsorption mechanism. Lastly, recovery and reuse of MMOFs were investigated in repeated adsorption cycles.

## 2. Results and Discussion

### 2.1. Synthetic Aspects

There are three important aspects under consideration when designing MMOFs:
Selecting the MNPs building blocks;Selecting the MOF framework;Selecting the design pattern.

In almost the full spectra of magnetic applications, ferrite MNPs stand on the top of pyramid and in our current work, doped ferrite MNPs [23,24,25,26,27] were chosen, both Co- and Zn-doped. Doping provides protection from undesired oxidation phenomena to which magnetite is susceptible (the iron-only ferrite that is more commonly investigated) that are more frequent than usual when meta-synthetic chemical procedures are employed for the MMOF fabrication. Furthermore, beneficial magnetic properties can be present in doped ferrites, such as zinc [24]. Regarding the chosen synthetic approach, autoclave route provides reproducible results, energy efficiency and isolation of coated MNPs that are readily able to form stable suspensions [23,24,25,26,27]. Herein, two diverse coatings were established, the octadecylamine (ODA) coating on the Co-doped ferrite MNPs and the PEG coating on the Zn-doped ferrite MNPs, both coatings previously employed by our group [28,29,30,31]. ODA and PEG when they are used in a triple role (solvent/surfactant/reducing agent) provide amine and hydroxyl groups respectively that extend from the surface of the MNPs, acting as donors that facilitate meta-synthetic architecting such as the attachment to MOF. Meanwhile, the coating prevents aggregation and therefore allows the particles to be more isolated, and a significant surface area is favored in this way. Thus, it provides the highly sought in meta-synthetic procedures colloidal stability that favors minimal decline in magnetic properties and SSAs of the MNA. As for the MOF outline, the well-known UiO-66, a high SSA MOF [32,33,34] (over 1300 m^2^/g), stable in aquatic solution was selected to be generated in-situ in the presence of the MNPs via a previously published facile route. A semi-self-assembly that resembles the so-called embedding approach was carefully selected amongst other reported strategies, such as simple mixing, layer-by-layer approach and/or encapsulation [14,15,16,17,18,19,20,21,22]. This choice aims to satisfy the bench-to-industry need for the few-in-number fabrication steps that layer-by-layer and encapsulation strategies lack, as both are complex and multistep syntheses. While simple mixing also satisfies the above-mentioned need it often fails to produce uniform and compact final structures, a goal that can also be achieved via the present semi-self-assembly procedure. Moreover, the affinity of the amine and hydroxyl of the MNPs to the Zr ions ensures stabilization of the two matrices (MNPs and MOF) through covalent and/or hydrogen bonding. To our knowledge, it is the first reported utilization of coated MNPs in the preparation of MMOFs.

### 2.2. Characterization of MMOFs

#### 2.2.1. CF MMOF

Figure 1 presents the main results derived by different characterization techniques for CF MMOF. SEM imaging at Figure 2 depicts the presence of polygonal particles with rough surfaces and sizes ranging from a few hundreds of nanometers to a few micrometers. SEM-EDS (Figure 1A and Appendix A with full EDS spectrum)reveal a composition of 82% *w*/*w* Zr and 18% *w*/*w* Fe. The Co-doped MNPs were also investigated by SEM-EDS (Appendix A) prior to the transformation to CF MMOF and have a composition of 66% *w*/*w* Fe and 34% *w*/*w* Co. Converting those values by dividing the % *w*/*w* Fe of CF MMOF with that of Co-doped MNPs resulted in a final MNPs content of 27% *w*/*w* for the CF MMOF. Moreover, the EDS spectra image portrays a uniform distribution of Zr and Fe in the CF MMOF with no particle segregation or aggregation apparent. Figure 1B illustrates the XRD graph of the CF MMOF where peaks match both the simulated pattern for the UiO-66 [32,33,34] and the cobalt ferrite pattern (ICDD-PDF #79-1744). Crystallite size for the MNPs and UiO-66 framework was estimated at 15 nm and 25 nm, respectively, via fitting the diffraction data and utilizing the Scherrer formula. The difference in the sizes is in accordance with the embedding approach. Figure 1C compares the FTIR spectra of CF MMOF, Co-doped MNPs and UiO-66 (synthesized in the absence of MNPs for comparative purposes). Peaks for both the Zr-O and the Fe-O appear in the CF MMOF spectra in the 800–500 cm^−^^1^ area. Full FTIR spectra with peaks indicated in the spectra are given in the Appendix A.

Furthermore, UiO-66 framework peaks dominate the CF MMOF spectra with shoulders of the ODA coating of Co-doped MNPs also being evident. Analogous comparative TGA curves (Figure 1D) verify the co-existence of MNPs and UiO-66 in the CF MMOF. In particular, the ODA coating forms a double layer around the MNPs, with the inner layer decomposing at elevated temperatures above 600 °C [25]. Meanwhile, free amines are provided as it has been shown before by the ninhydrin colorimetric assay [30]. In the TGA curve of CF MMOF, up to 600 °C, the curve resembles that of UiO-66 while above 600 °C it is more similar to that of the inner ODA layer of Co-doped MNPs. This can be attributed to an explicit conjugation of the two matrices, where the outer layer of ODA is transformed via the process. This also explains the faint ODA shoulders in the FTIR of CF MMOF.

#### 2.2.2. ZF(acac) MMOF

Two different ZF MMOFs were fabricated by varying the zinc precursor during the synthesis of Zn-doped MNPs. This strategy has been previously employed by our group to diversify the zinc doping [24]. Figure 3 demonstrates the characterization measurements for ZF(acac) MMOF. Irregular-shaped particles with coarse surface and polygonal particles with sizes in the range from a few hundreds of nanometers up to a few micrometers are spotted in the SEM image of Figure 4. SEM-EDS compositions for Zn-doped(acac) MNPs (Appendix A) and ZF(acac) MMOF (Figure 2A and Appendix A with full EDS spectrum) were found at 71% *w*/*w* Fe and 29% *w*/*w* Zn and 82% *w*/*w* Zr with18% *w*/*w* Fe, respectively, giving a final MNPs content of 25% *w*/*w* for the ZF(acac) MMOF. EDS spectrum image reveals a homogeneous elemental distribution of Zr and Fe, indicating a rather undisturbed in-situ growth pattern of the MOF around the primary ferrite nanoparticles. XRD diffractogram of ZF(acac) MMOF (Figure 3B) contains peaks corresponding to both the simulated pattern for the UiO-66 and the zinc ferrite pattern (ICDD-PDF #74-2397). Crystallite size for the MNPs and UiO-66 framework was estimated at 20 nm and 25 nm, respectively. Comparative FTIR spectra (Figure 3C) and TGA curves (Figure 3D) provide similar results to that of CF MMOF. Full FTIR spectra with peaks indicated in the spectra are given in the Appendix A. In detail, PEG also forms a double layer around the MNPs decomposing at even higher temperatures above 700 °C [35] and the inner layer is present in the ZF(acac) MMOF as indicated by the TGA recordings. The outer layer is possibly dissolved as suggested by the faint shoulders in the FTIR spectra while both the Zr-O and the Fe-O are present in the ZF(acac) MMOF spectra in the 800–500 cm^−^^1^ area.

#### 2.2.3. ZF(Cl) MMOF

Figure 5 displays the main characterization results for ZF(Cl) MMOF.SEM imaging at Figure 6 illustrates the prevalence of polygonal particles with size in the microscale. Though many of the particles seem to have a somewhat rough surface, a good portion of this sample appears to contain smoother particle surfaces and more clear (less blurred) edges and corners compared to the previous samples. Compositions for Zn-doped(acac) MNPs (Appendix A) and ZF(Cl) MMOF (Figure 3A and Appendix A with full EDS spectrum) were found at 95% *w*/*w* Fe and 5% *w*/*w* Zn and 65% *w*/*w* Zr with 35% *w*/*w* Fe, respectively, providing a final MNPs content of 37% *w*/*w* for the ZF(Cl) MMOF. EDS spectra image illustrates an even distribution of Zr and Fe throughout the whole zone tested in the microscale size regime. By utilizing the ZnCl_2_ precursor, the regulation of the zinc doping was achieved according to the polyol process mechanism and based on the stability of intermediate complexes formed at an initial stage, as previously shown by our group [24]. Additionally, the content of MNPs in ZF(Cl) MMOF is found significantly higher than the CF and ZF(acac) MMOFs. XRD peaks of ZF(Cl) MMOF (Figure 5B) match both the simulated pattern for the UiO-66 and the zinc ferrite pattern (ICDD-PDF #74-2397). Crystallite size for the MNPs and UiO-66 framework was estimated at 13 nm and 25 nm, correspondingly. This is the biggest size difference amongst the three MMOFs and is possibly the crucial factor for the significantly higher content of MNPs in ZF(Cl) MMOF. Comparative FTIR spectra (Figure 5C) and TGA curves (Figure 5D) for ZF(Cl) MMOF offer similar insights to what was discussed above regarding the ZF(acac) MMOF. Full FTIR spectra with peaks indicated in the spectra are given in the Appendix A.

### 2.3. Magnetic Properties and SSA Values

VSM hysteresis loops for the three MMOFs can be observed in Figure 7. Saturation magnetization was measured at 38, 42 and 22 emu/g for CF, ZF(acac) and ZF(Cl) MMOFs, respectively, while coercivity values derived were 300, 130 and 130 Oe for CF, ZF(acac) and ZF(Cl) MMOFs, correspondingly. Higher coercivity was indeed expected for CF MMOF due to the Co doping [25]. Furthermore, M-H plots were also recorded for the MNPs building blocks (Appendix A) to estimate the magnetization quenching and saturation magnetization which were found at 101, 129 and 117 emu/g for Co-doped, Zn(acac)-doped, Zn(Cl)-doped MNPS, respectively. Zinc doping has been shown to induce an increase in magnetization due to favorable spin-spin interactions in the spinel lattice [24]. However, based on reported studies, [24] the low Zn-doping in the Zn(Cl)-doped MNPs should lead to the highest magnetization value amongst the two Zn-doped MNPs and that is not the case here. This can be explained by considering the crystallite sizes. As Zn(acac)-doped MNPs are significantly larger, size effects (larger size results in larger magnetization) outshine doping effects. Magnetization decrease values, stemming from ratio of ([Ms of MMOFs]/[Ms of MNPs]) × 100 are calculated at 62%, 67% and 81% for CF, ZF(acac) and ZF(Cl) MMOFs, respectively. Regarding comparison with the saturation magnetization values of MMOFs in the literature, Refs. [19,20,21,22] they span in the range of 3 to 73 emu/g, making the saturation magnetization values of the current MMOFs higher than about 50% of the reported MMOFs. Low saturation magnetization values of MMOFs occur mainly because of the spacer/organic coating that is added to cover bare MNPs, for instance silica coated Fe_3_O_4_ nanoparticles resulted to UiO-66@Fe_3_O_4_@SiO_2_ with magnetization value of 8.1 emu g^−w^ [36]. Controlling the thickness and/or the coating percentage onto MNPs is deemed as crucial to preserve high Ms magnitudes in nanocomposites [8,31]. This is particularly important as first-rate magnetic properties ensure the recycle-reuse modality with the use of common/commercial magnets, that is among the primary desired functions of the MMOFs.

N_2_ absorption isotherms for all three MMOFs as well as UiO-66 are presented in Figure 8. SSAs are calculated via this isotherms at 1132, 332, 244 and 311 m^2^/g for UiO-66, CF MMOF, ZF(acac) MMOF and ZF(Cl) MMOF, respectively. SSA reduction is estimated at 71%, 78% and 71% for CF, ZF(acac) and ZF(Cl) MMOFs correspondingly when compared with the initial MOF (UiO-66). Additionally, the integration of the two matrices (MNPs and MOF) is indicated and ensures high stability. However, the resulted SSA magnitudes can be considered moderate and higher than about 25% of the reported MMOFs that span in the range of 17 to 1248 emu/g [19,20,21,22]. Table 1 sums the physicochemical traits for all three MMOFs.

### 2.4. Pesticide Adsorption Studies

2,Dand 2,4,5-T have been used as pesticides for over 70 years [37]. However, health risks are associated with both of them, and it is considered vital to ensure that their quantity in drinking, domestic and irrigation waters is below the actual health risk limits. For that purpose, adsorption studies for all three MMOFs were carried out in varying pesticide concentrations (10, 20, 40, 80, 120, 160, 200, 250 ppm) at a steady time frame of 48 h. After the end of the experiments, MMOFs were removed from the solution via the use of external commercial magnet and the remaining pesticide concertation was identified via UV-Vis. Data were fitted to Langmuir and Freundlich models [38,39,40,41,42,43,44] for both linear and non-linear fitting and results are summarized in Table 2. The R^2^ values for the Langmuir model are higher than the Freundlich ones, suggesting that the adsorption action of MMOFs is characterized by a maximum adsorption value (Q_max_) and the chemisorption took place on a monolayer on a homogeneous surface containing identical active sites [45]. Both the linear and the non-linear fitting of the Langmuir model led to high R^2^ values. Q_max_ values were plotted for all three MMOF and both pesticides and are demonstrated in Figure 9. CF MMOF and ZF(Cl) MMOF displayed higher Q_max_ values than ZF(acac) MMOF, in correlation with their SSA. Furthermore, all three MMOF exhibited higher adsorption capacity for the 2,4-D possibly due to the hindrance caused by the extra Cl- atom of the 2,4,5-T.

Kinetic studies [46,47,48,49] for all three MMOFs and both pesticides were also performed at a fixed pesticide concentration (200 ppm) at time intervals of 2, 4, 8, 24, 48, 72, 96, 144 and 192 h. Data were fitted to pseudo first order (PFO) and pseudo second order (PSO) models, both linearly and non-linearly. PFO resulted in poor R^2^ values and is rejected. Table 3 summarizes the PSO data. PSO better fit indicates a fast adsorption rate in the beginning that slows down over time.

Lastly, after their adsorption studies, all three MMOFs were collected, washed, dried and reused in new adsorption cycles using a stock solution of 200 pm 2,4-D pesticide. The goal of this study was to investigate the efficiency of their recovery and reusability potential. Figure 10 illustrates the UV-Vis curves in the vicinity of the 2,4-D characteristic absorbance, after 24 h of adsorption period for all three washed and dried MMOFs, in comparison with the 2,4-D starting solution curve. ZF(Cl) MMOF displayed by far the best potential amongst the three while it is worth mentioning that all samples are considered suitable for recovery and reuse. Although the sample under discussion seemed to contain also some smooth-shaped particles, its good percentage of particles with uneven surfaces probably favored the trapping and removing of pollutants.

Considering the adsorption mechanism is still unclear to us as further enlightening experiments are needed. However, formerly four possibilities have been suggested: (i) coordination in unsaturated metal centers of MMOFs, (ii) π-π stacking interactions via the organic rings of MOF, (iii) electrostatic interactions, and (iv) bonding through functional groups of MMOF surface [45,50]. At present, the (iv)case is considered unlikely as NH_2_ and/or OH functional groups of MNP surfaces are bonded in the integrated matrix MMOF based on TGA and FTIR evidence. Electrostatic interactions are also excluded as both pesticides are negative-charged. Given the similar structures and the presence of phenyl ring in both pesticides we assume π-π stacking interactions via the organic rings of UiO-66 framework as the most possible interaction while adsorption onto coordinatively unsaturated sites cannot be excluded.

## 3. Materials and Methods

### 3.1. Materials 

Iron(III)acetylacetonate Fe(acac)_3_(Merck-Schuchardt, M = 353.18 g mol^−^^1^), cobalt(III)acetylacetonate Co(acac)_3_(Merck-Schuchardt, M = 356.26 g mol^−^^1^), octadecylamine(ODA; Sigma-Aldrich, 90%, M = 269.509 g mol^−^^1^), zinc(II)acetylacetonate hydrate Zn(acac)_2_(Sigma-Aldrich, M = 263.61 g mol^−^^1^), zinc chloride dehydrate ZnCl_2_(Sigma-Aldrich, 90%, M = 269.509 g mol^−^^1^), polyethylene glycol (PEG; Alfa Aesar, M = 8000 g mol^−^^1^), zirconium(IV) chloride dehydrate ZrCl_4_(J&K Scientific, 98%, M = 233.04 g mol^−^^1^), terephthalic acid (H_2_BDC; J&K Scientific, 99%, M = 166.13 g mol^−^^1^), 2,4-dichlorophenoxyacetic acid (2,4-D; Sigma, Minimum 98%, M = 221.04 g mol^−^^1^), 2,4,5-trichlorophenoxyacetic acid (2,4,5-T; Sigma, Approx. 97%, M = 255.49 g mol^−^^1^).

### 3.2. Synthesis of MNPs Building Blocks

Three kinds of coated doped ferrite MNPs were prepared via similar autoclave routes. ODA-coated Co-doped MNPs were produced in an autoclave by the decomposition of acetylacetonate iron(III) and cobalt(III) at a 2:1 ratio, Fe(acac)_3_ 1.8 mmol/Co(acac)_3_ 0.9 mmol in the presence solely of ODA 12.9 mmol. Accordingly, PEG-coated Zn-doped MNPs were fabricated by the decomposition of acetylacetonate iron(III) and the zinc precursor, being zinc acetylacetonate(II) and zinc chloride(II) for the two diverse syntheses at a 2:1 ratio, Fe(acac)_3_ 1.8 mmol/Zn(acac)_2_ or ZnCl_2_ 0.9 mmol in the presence solely of PEG 9.375 mmol. In all instances the temperature of the oven was elevated with a steady rate (4 °C/min) to 200 °C and was kept stable for 24 h. After the 24 h reaction autoclaves were left to cool down to room temperature with a rate of 5 °C/min. Co- and Zn-doped ferrite MNPs (hereafter named as Zn-doped(acac) MNPs and Zn-(doped)Cl MNPs, respectively) were isolated after repeating washing cycles with EtOH and centrifugations (5000 rpm), followed by solvent evaporation. 

### 3.3. Synthesis of UiO-66

UiO-66 was prepared via a previously reported facile route according to which ZrCl_4_ (0.108 mmol) and benzene-dicarboxylic acid (H_2_BDC; 0.15 mmol) were mixed and dissolved in DMF (3 mL) and HCl (0.2 mL) [32]. Specifically, a vial was loaded with ZrCl_4_, one third of the DMF, and concentrated HCl before being sonicated for 20 min until fully dissolved. H_2_BDC and the remaining portion of the DMF were then added and the mixture was sonicated for another 20 min (without the ligand being completely soluble under these conditions though) before being heated at 80 °C overnight in a conventional oven. After the spontaneous cooling of the vessel to room temperature its content was transferred to a 15 mL falcon and a white fine-grained solid was isolated through washing cycles with DMF and EtOH and centrifugations (5000 rpm), followed by solvent evaporation.

### 3.4. Fabrication of MMOFs

Three different MMOF materials originating from Co-doped, Zn-doped(acac) and Zn-doped(Cl) MNPs (hereafter named as CF MMOF, ZF(acac) MMOF and ZF(Cl) MMOF, respectively) were synthesized by using the following general procedure. Briefly, 0.025 g of the corresponding ferrite MNPs were dispersed in DMF (29 mL) in a two-neck round bottom flask before the addition of ZrCl_4_ (0.060 g) and H_2_BDC (0.067 g) and subsequent 20-min sonication (the ligand was not completely soluble). UiO-66 growth was promoted through a heat-driven reaction under reflux conditions and simultaneous mechanical stirring for 12 h under a stable temperature of 120 °C. After the reaction, the vessel was left to cool down to room temperature while the magnetic material was collected through the application of external commercial magnet before the removal of the supernatant. Repeated washing cycles and centrifugations (5000 rpm) with DMF and EtOH and subsequent solvent evaporation yielded powder-like magnetic solids, which were left to dry under vacuum.

### 3.5. Characterization

The elemental composition of the samples was tested by field emission gun scanning electron microscopy (FEG-SEM; Tescan Lyra dual beam microscope), where energy dispersive spectrometry (EDS; X-Man^N^) spectra were obtained. A 20 mm^2^ SDD detector (Oxford Instruments) and Aztec Energy software were used. Samples for SEM were placed on a carbon conductive tape. For SEM and SEM-EDS measurements, a 10 kV electron beam was employed. The crystal structures of the MMOFs were investigated through X-ray diffraction (XRD; Bruker D8-Bragg-Brentano para focusing geometry diffractometer) performed in the 2*θ* region from 5 to 90°, with monochromatized Cu K*a* radiation (*λ* = 1.5418 Å). Fourier transform infrared spectroscopy (FTIR; Nicolet FTIR 6700 spectrometer, 450–4000 cm^−^^1^) was recorded with samples prepared as KBr pellets. Thermogravimetric analysis (TGA; SETA-RAM SetSys-1200 instrument) was performed at a heating rate of 10 min^−^^1^ under N_2_ atmosphere in the range of ambient temperature to 900 °C. Magnetic measurements were acquired by a vibrating sample magnetometer (VSM; Oxford Instruments 1.2 H/CF/HT VSM). The specific surface area of the as-prepared MMOFs samples was measured via N_2_ porosimetry, using the Brunauer-Emmett-Teller theory (BET; Tristar 3000 Micrometrics). Degassing was carried out using N_2_ gas at 110 °C for 2 h, cryogenic conditions were achieved with liquid nitrogen while helium was also employed as a purging gas. The absorbance measurements of all pesticide solutions were performed on a double beam ultraviolet-visible spectrometer (UV-Vis; Hitachi U-2001).

### 3.6. Adsorption Studies

Calibration curves of 2,4-D and 2,4,5-T were prepared by measuring the absorbance at 283 and 287.5 nm via UV-Vis spectrometry, respectively, with a series of standard pesticide aqueous solutions (0–250 ppm). The adsorption experiments were executed at ambient temperature for a series of pesticide solutions with varying concentrations. Before adsorption, the adsorbents were dried under vacuum overnight. The pesticide solutions were prepared by dilutions of a stock solution (250 ppm) with deionized water and the pH of the final solutions was adjusted to 3.5 by the addition of minimum volume of HCl. For each concentration, 10 mg of ground MMOF sample was added to 50 mL of the pesticide solution in a conical flask and stirred for 48 h at 130 rpm. After adsorption, the adsorbent was magnetically separated with the use of external magnet and the remaining amount of pesticide in the solution was monitored by UV-Vis spectrometry by using the prepared calibration curves based on Lambert-Beer law.

### 3.7. Kinetic Studies

For the kinetic studies, adsorption experiments were conducted at ambient temperature in conical flasks containing 50 mL of 2,4-D solution (200 ppm) or 2,4,5-T solution (80 ppm) of pH 3.5. 10 mg of adsorbent were added to each solution, which was stirred with shaking speed of 130 rpm. 2 mL aliquots were taken with pipette at certain time intervals after magnetic separation of the adsorbent with external magnet on the walls of the vessel, while it was temporarily removed from the stirrer. The concentration of the contaminants in the aliquots was measured by UV-Vis spectrometry. The sampling of aliquots was only terminated after the adsorption process was considered to have reached a dynamic equilibrium.

### 3.8. Recovery and Reuse

Adsorbed MMOFs with loaded 2,4-D were treated with repeating washing cycles with deionized water and ethanolic solutions by vortex shaking to release the pesticide. After the evaporation of the solvent the adsorbents were dried under vacuum overnight and were tested in subsequent adsorption cycles by the addition of 10 mg in newly prepared 2,4-D solution (200 ppm) and the measurement of the remaining amount of pesticide after 24 h of contact.

## 4. Conclusions

The motivation to use MMOFs for water pollution remediation is rising. However, for practical usage the synthetic procedure is needed to not be complicated as the cost, safety and convenience are required parameters. In that vein, three novel MMOFs have been prepared through a facilitated procedure by (i) the employment of a well-studied MOF such as UiO-66 with exceptional stability and reproducibility and without any additional functionality and (ii) well established, stable, and reproducible coated MNPs. The resulting MMOFs presented a roughly 70% reduction in both the magnetic properties and the SSA in comparison with their building blocks; nevertheless, the remaining 30% of magnetization and SSA was adequate for successful and sufficient adsorption of two pesticides that constitute health risks. As the properties of the UiO-66 framework remained fixed in all three materials, differences in performance are attributed to the composition of the MNPs. Based on that, MNPs with the smallest crystallite size as well as a PEGylated surface achieved the highest MNP loading in the framework in combination with the smallest SSA reduction and performed better as adsorbents. Comparatively, to our knowledge, the present MMOFs displayed higher saturation magnetization values than 50% and higher SSA values than 25% of the reported MMOFs. Meanwhile, they can be recovered and reused via commercial magnets to satisfy the need for sustainable and low-cost innovation.

## Figures and Tables

**Figure 1 molecules-28-00039-f001:**
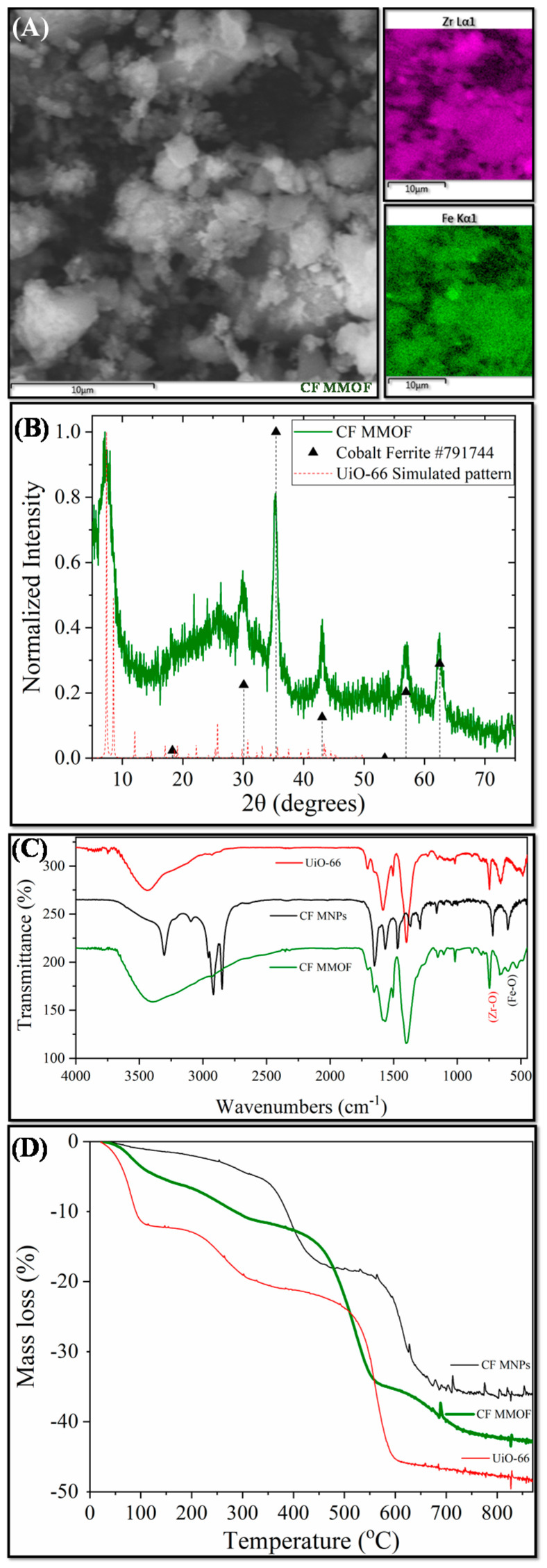
(**A**) SEM image; (**B**) XRD measurement; (**C**) FTIR spectrum; and (**D**) TGA of CF MMOF.

**Figure 2 molecules-28-00039-f002:**
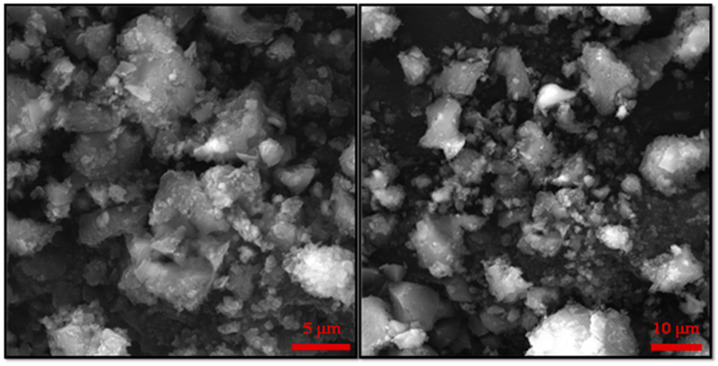
Additional, higher resolution SEM images for the sample CF MMOF.

**Figure 3 molecules-28-00039-f003:**
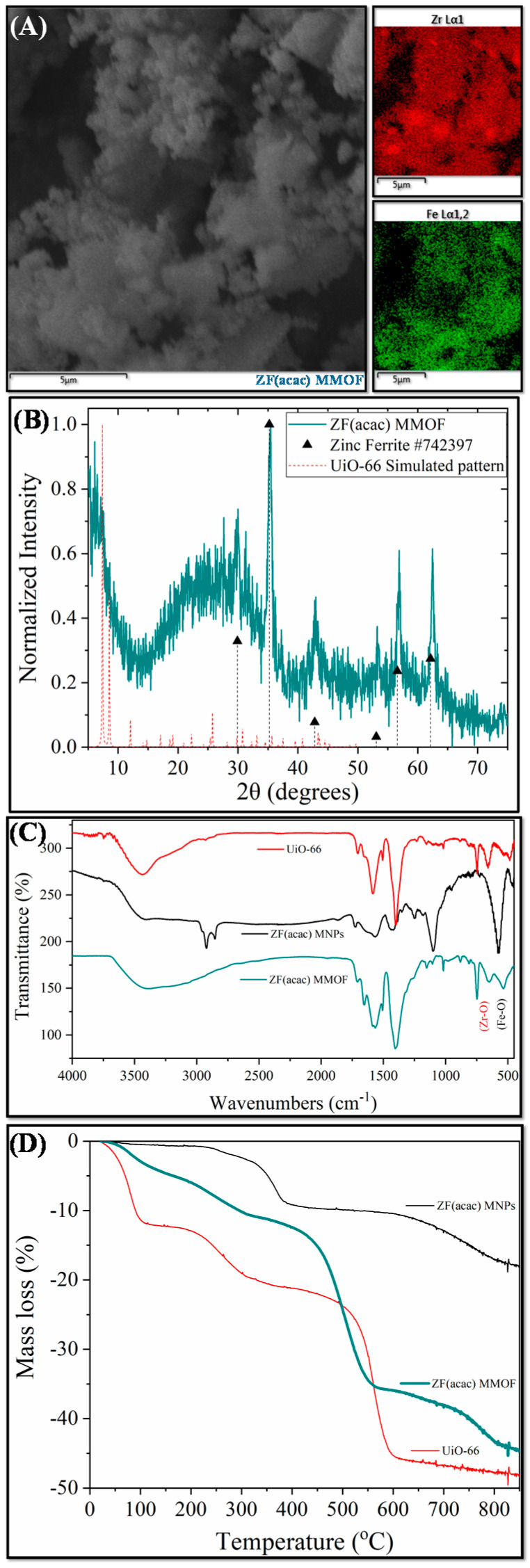
(**A**) SEM image; (**B**) XRD measurement; (**C**) FTIR spectrum; and (**D**) TGA of ZF(acac) MMOF.

**Figure 4 molecules-28-00039-f004:**
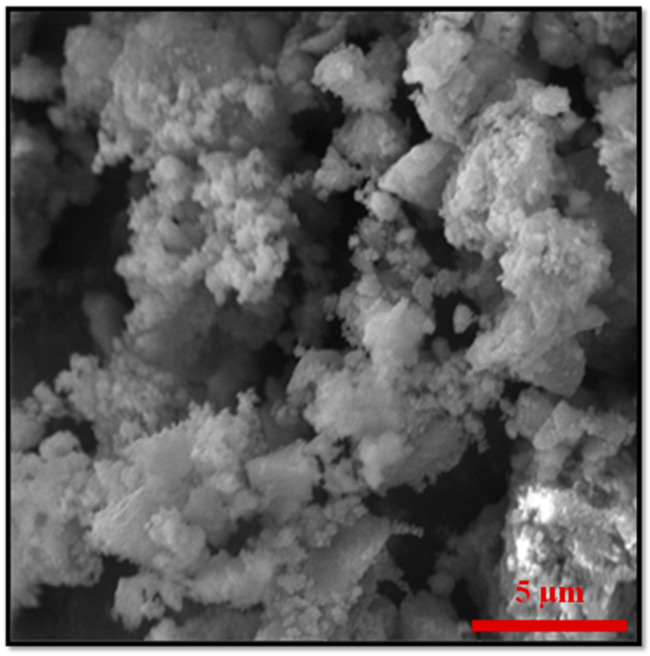
Additional, higher resolution SEM image for the sample ZF(acac) MMOF.

**Figure 5 molecules-28-00039-f005:**
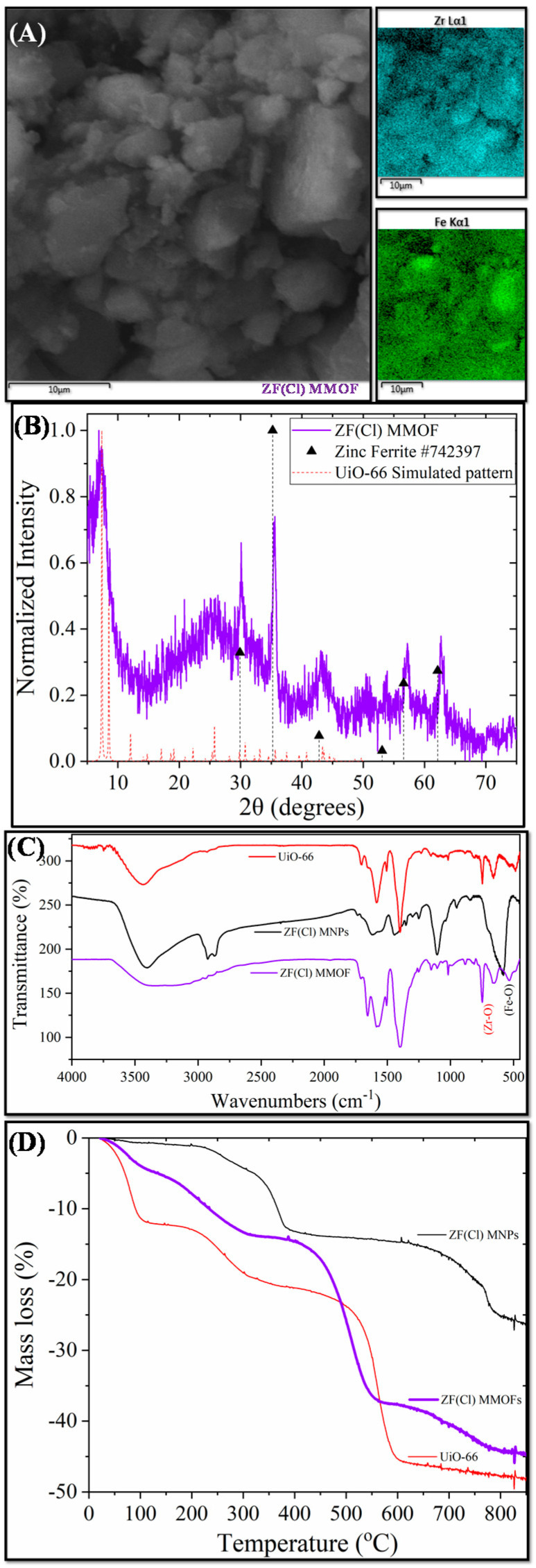
(**A**) SEM image; (**B**) XRD measurement; (**C**) FTIR spectrum; and (**D**) TGA of ZF(Cl) MMOF.

**Figure 6 molecules-28-00039-f006:**
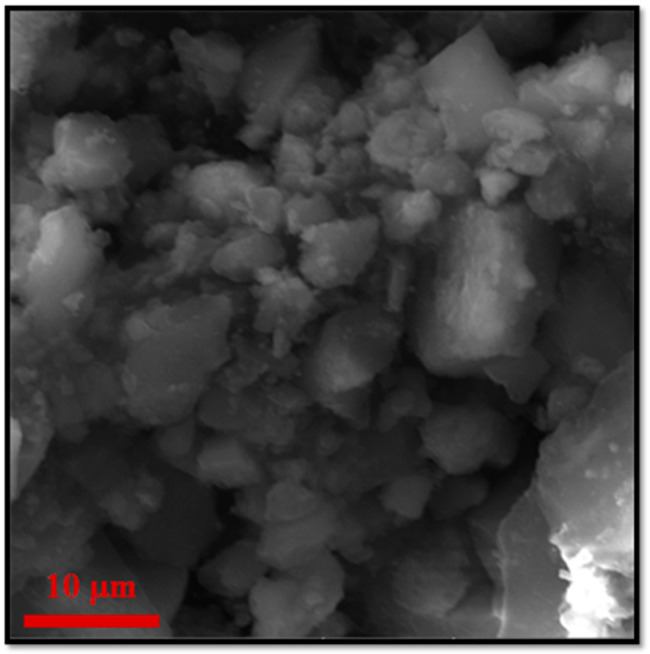
Additional, higher resolution SEM image for the sample ZF (Cl) MMOF.

**Figure 7 molecules-28-00039-f007:**
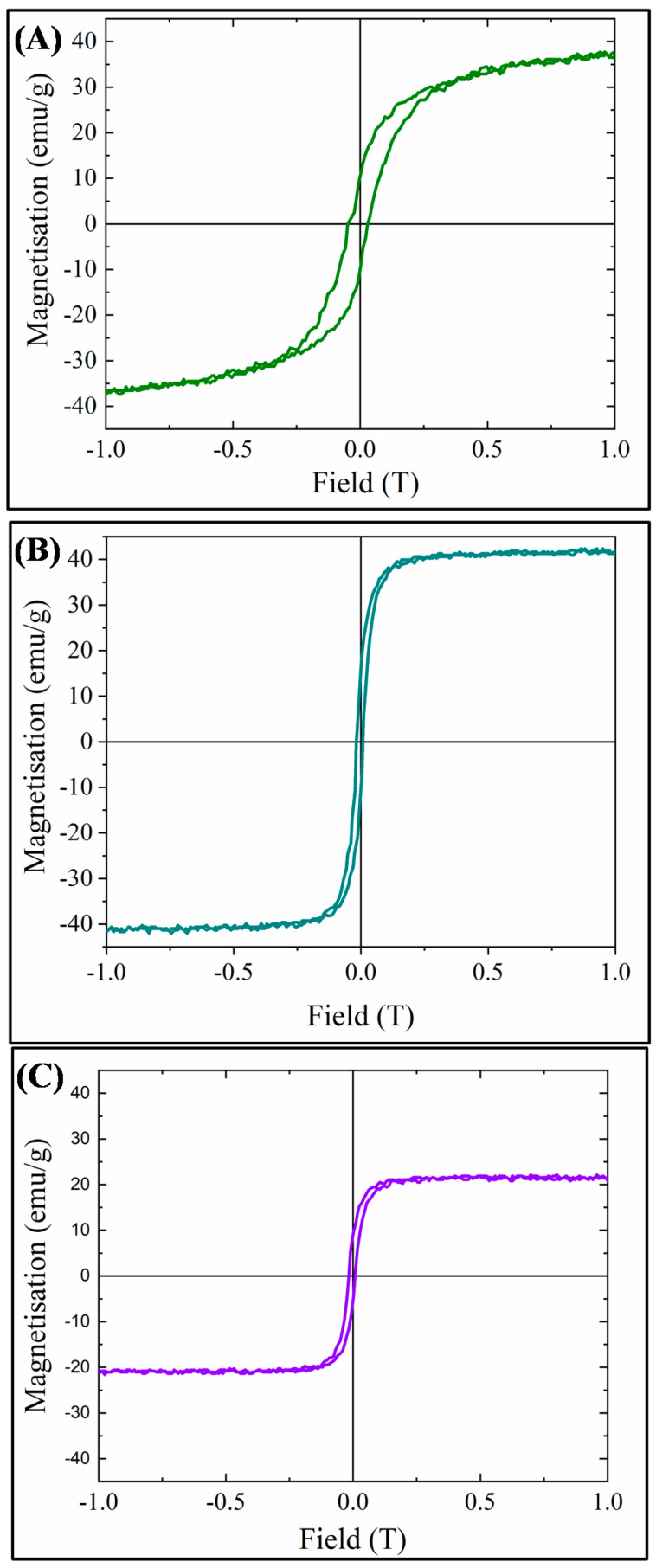
VSM hysteresis loops for: (**A**) CF MMOF; (**B**) ZF(acac) MMOF; and (**C**) ZF(Cl) MMOF.

**Figure 8 molecules-28-00039-f008:**
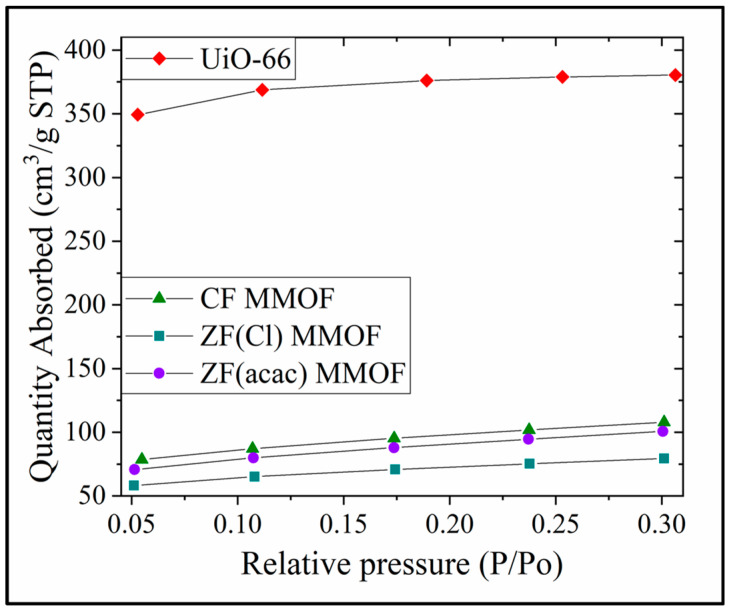
N_2_ Absorption isotherms for UiO-66, CF MMOF, ZF(acac) MMOF and ZF(Cl) MMOF.

**Figure 9 molecules-28-00039-f009:**
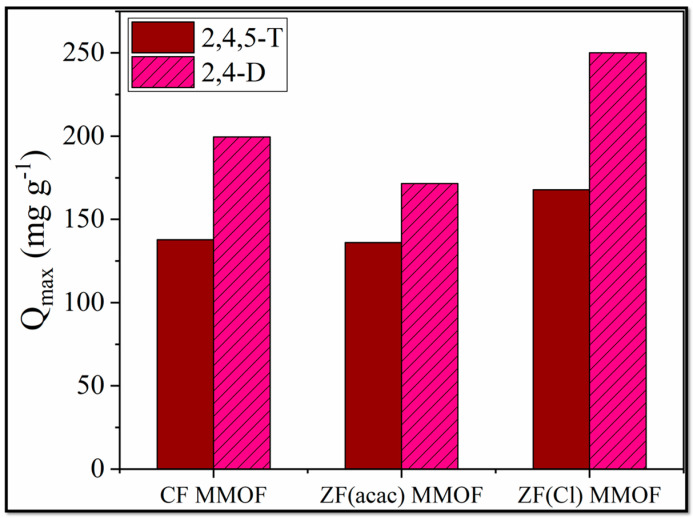
Q_max_ values for CF MMOF, ZF(acac) MMOF and ZF(Cl) MMOF for 2,4-D and 2,4,5-T pesticides.

**Figure 10 molecules-28-00039-f010:**
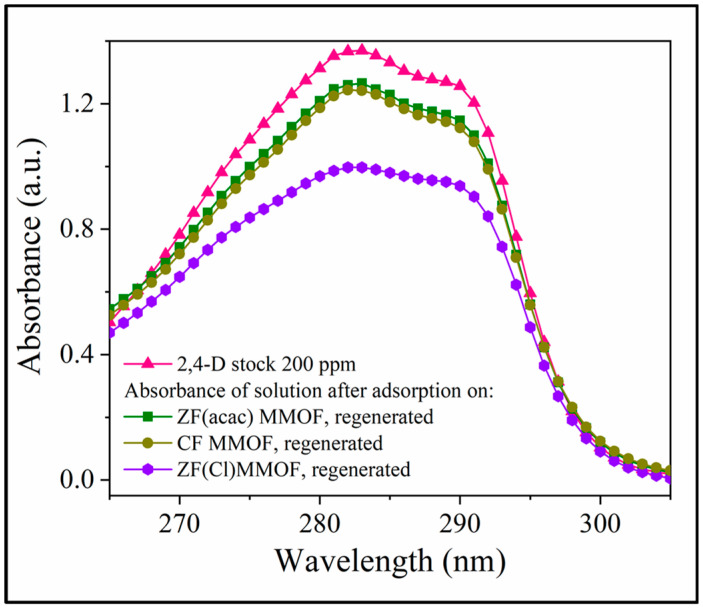
UV-Vis spectra after 24 h of adsorption period for all three washed and dried MMOFs, in comparison with the 2,4-D starting solution curve.

**Table 1 molecules-28-00039-t001:** MMOFs physicochemical traits.

Properties/MMOFs	CF	ZF(acac)	ZF(Cl)
MNPs Content (% *w*/*w*)	27	25	37
MNPs crystallite size (nm)	15	20	13
UiO-66 Framework crystallite size (nm)	25	25	25
Saturation Magnetization (emu/g)	38	42	22
Magnetization reduction (%)	62	67	81
SSA (m^2^/g)	332	244	311
SSA reduction (%)	71	78	71

**Table 2 molecules-28-00039-t002:** Adsorption data for all three MMOFs, both tested pesticides, fitted to Langmuir and Freundlich models, both linearly and non-linearly.

Isotherm	Pesticide	MMOF						
			CF MMOF		ZF(acac) MOF		ZF(Cl) MMOF	
			Fitting					
			Linear	Non-Linear	Linear	Non-Linear	Linear	Non-Linear
**Langmuir**	**2,4,5-T**	** *Q* ** ** _max_ **	137.74	140.62	136.05	138.75	167.79	174.37
		**K_L_**	0.02	0.02	0.02	0.02	0.05	0.05
		**R_L_**	0.19	0.20	0.23	0.03	0.11	0.12
		**Adj. *R*^2^**	0.963	0.943	0.972	0.969	0.982	0.944
	**2,4-D**	** *Q* ** ** _max_ **	199.60	174.66	171.53	174.66	250.00	257.84
		**K_L_**	0.07	0.07	0.08	0.07	0.02	0.02
		**R_L_**	0.07	0.08	0.07	0.08	0.21	0.22
		**Adj. *R*^2^**	0.943	0.837	0.988	0.967	0.957	0.948
**Freundlich**	**2,4,5-T**	**1/*n***	0.508	0.422	0.493	0.450	0.375	0.299
		**K_F_**	9.22	13.66	8.93	10.90	25.29	35.28
		**Adj. *R*^2^**	0.920	0.868	0.966	0.938	0.848	0.813
	**2,4-D**	**1/*n***	0.392	0.292	0.299	0.268	0.524	0.459
		**K_F_**	29.86	46.62	37.78	43.35	13.65	18.50
		**Adj. *R*^2^**	0.771	0.666	0.931	0.905	0.955	0.902

**Table 3 molecules-28-00039-t003:** Kinetic data for all three MMOFs, both tested pesticides, fitted to PSO model, both linearly and non-linearly.

Kinetic Model	Pesticide	MMOF						
			CF MMOF		ZF(acac) MMOF		ZF(Cl) MMOF	
			Fitting					
			Linear	Non-Linear	Linear	Non-Linear	Linear	Non-Linear
**PSO**	**2,4,5-T**	**q_e_**	137.17	140.72	115.47	118.17	137.36	139.48
		**K_2_**	2.17 × 10^−5^	1.65 × 10^−5^	2.14 × 10^−5^	1.67 × 10^−5^	1.07 × 10^−5^	9.49 × 10^−6^
		**H**	0.409	0.326	0.285	0.228	0.202	0.185
		**Adj. *R*^2^**	0.998	0.990	0.994	0.966	0.994	0.965
	**2,4-D**	**q_e_**	213.22	208.41	212.77	214.87	196.85	201.90
		**K_2_**	1.70 × 10^−5^	2.78 × 10^−5^	3.63 × 10^−5^	3.05 × 10^−5^	2.95 × 10^−5^	2.12 × 10^−5^
		**H**	0.775	1.208	1.645	1.409	1.144	0.866
		**Adj. *R*^2^**	0.989	0.940	0.995	0.978	0.995	0.977

## Data Availability

Not applicable.

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
