# Peer review of "Magnetic Nanocomposites of Coated Ferrites/MOF as Pesticide Adsorbents"

_molecules, 2022, doi:10.3390/molecules28010039_

Round 1

Reviewer 1 Report

In this work, Catherine Dendrinou-Samara et al. have presented the use of coated ferrite magnetic nanoparticles (MNPs) in the synthesis of magnetic UiO-66 metal-organic framework (MMOFs) composites. They indicated that coated MNPs facilitated the conjugation and stands as a novel strategy for fabricating MMOFs with increased stability and explicit structure. In addition, they disclosed that roughly 70% reduction was observed in magnetic properties and specific surface area compared with initial MNPs building blocks and UiO-66 framework and the remaining 30% was adequate for successful and sufficient adsorption of two different pesticides. However, minor revisions are required from authors to finally accept this manuscript as follows.

(B) Scientific issues

(1) In this paper, the authors claimed that using coated magnetic nanoparticles is presented for the first time; however, by conducting literature survey, a previous work presented the use of silica-coated Fe3O4 nanoparticles to prepare magnetic UiO-66@Fe3O4@SiO2 catalyst for the catalytic synthesis of cyclic carbonates from epoxides and carbon dioxide [Appl Organometal Chem. 2017; 31:e3656. https://doi.org/10.1002/aoc.3656]. The authors should compare the silica report with the current work under submission to show differences between both coated materials.

(2) The authors should declare why coated ferrite enhanced magnetization and surface area of the resultant magnetic composite.

(3) The title should be “Magnetic nanocomposites of coated ferrites/MOF as pesticide adsorbents” by deleting the word “formation

(A) Language issues

The manuscript is written at high level; however, it requires fixation for some issues as follows:

(1) There should be no space between % and number (i.e., 90% not 90 %)

(2) oC is written in wrong way; it should be ℃

(3) Title, headings, and subheadings should not be ended with period.

(4) Metal organic frameworks should be hyphenated as metal-organic frameworks

(5) Use consistent hyphenation for semi self assembly

The authors should do editing and proofreading to this manuscript to provide the best writing style free of errors.

By addressing the above issues, this manuscript is eligible for publication in the journal of “Molecules”.

Author Response

Reviewer 1

In this work, Catherine Dendrinou-Samara et al. have presented the use of coated ferrite magnetic nanoparticles (MNPs) in the synthesis of magnetic UiO-66 metal-organic framework (MMOFs) composites. They indicated that coated MNPs facilitated the conjugation and stands as a novel strategy for fabricating MMOFs with increased stability and explicit structure. In addition, they disclosed that roughly 70% reduction was observed in magnetic properties and specific surface area compared with initial MNPs building blocks and UiO-66 framework and the remaining 30% was adequate for successful and sufficient adsorption of two different pesticides. However, minor revisions are required from authors to finally accept this manuscript as follows.

(B) Scientific issues    

(1) In this paper, the authors claimed that using coated magnetic nanoparticles is presented for the first time; however, by conducting literature survey, a previous work presented the use of silica-coated Fe3O4 nanoparticles to prepare magnetic UiO-66@Fe3O4@SiO2 catalyst for the catalytic synthesis of cyclic carbonates from epoxides and carbon dioxide [Appl Organometal Chem. 2017; 31:e3656. https://doi.org/10.1002/aoc.3656]. The authors should compare the silica report with the current work under submission to show differences between both coated materials.

Our response: We thank the reviewer for the suggested comparative discussion. A segment has been added in the revised manuscript, page 9:

‘Low saturation magnetization values of MMOFs occur mainly because of the spacer/organic coating that is added to cover bare MNPs, for instance silica coated Fe3O4 nanoparticles resulted to UiO-66@Fe3O4@SiO2 with magnetization value of 8.1 emu g−1. [36] The thickness and/or the coating percentage onto MNPs is regarding essential to preserve high Ms magnitudes in nanocomposites.[8,31] ’

(2) The authors should declare why coated ferrite enhanced magnetization and surface area of the resultant magnetic composite.

Our response: As it has been stated (Results and discussion section, page 3) ODA and PEG when they are used in a triple role (solvent/surfactant/reducing agent) provide amine and hydroxyl groups respectively that extend from the surface of the MNPs, acting as donors that facilitate metasynthetic architecting such as the attachment to MOF. Additionally, the coating prevents aggregation and thus allows the particles to be more isolated, thus having a significant surface area(aggregated particles have smaller surface area). To clarify, in the revised manuscript have been added:

At page 3: 'Meanwhile, the coating prevents aggregation and therefore allows the particles to be more isolated, and a significant surface area is endowing. Thus, it provides the highly sought in metasynthetic procedures colloidal stability that favors minimal diminish in magnetic properties and SSAs of the MNA.’

At page 5: 'Meanwhile, free amines are provided as it has been shown before by the ninhydrin colorimetric assay.[30]'

(3) The title should be “Magnetic nanocomposites of coated ferrites/MOF as pesticide adsorbents” by deleting the word “formation”

Our response: We thank the reviewer for the suggestion. Title has been revised, accordingly.

(A) Language issues

The manuscript is written at high level; however, it requires fixation for some issues as follows:

(1) There should be no space between % and number (i.e., 90% not 90 %)

(2) oC is written in wrong way; it should be ℃

(3) Title, headings, and subheadings should not be ended with period.

(4) Metal organic frameworks should be hyphenated as metal-organic frameworks

(5) Use consistent hyphenation for semi self assembly

The authors should do editing and proofreading to this manuscript to provide the best writing style free of errors.

Our response: We thank the reviewer, all the above issues have been addressed in the revised manuscript.

Reviewer 2 Report

1. Please mark the main peaks of the various functional groups in Fig. 1C.

2. From the Fig. 2, the sample are not impurity. Why?

3. Please provide the recycle data.

4. In the introduction, the authors have stated “On the other hand, conventional adsorbents in the likes of zeolites [11, 12] 45 and/or novel engineered materials such as metal organic frameworks (MOFs) [13-15]” The MOFs have been extensively studied by scientists and engineers as a fascinating feature the exploration of their use in numerous scientific domains, such as energy storages, environmental pollution, sensing platforms, and catalysis, photocatalysis, oxidation, hydrogenation due to their diverse active sites. Some more updated refs may be considered, such as Micropor. Mesopor. Mat, 341(2022) 112098; Mater. Today. Commum., 2022, 31,103514; and Dalton Trans., 2021, 50, 18016–18026; Inorganics, 10(2022) 202; Chem. Eng. J., 2022, 433, 133857.

5. The particles sizes for different samples should be counted.

6. What about the precise content of Co and carbon species in the materials?

7. Since pharmaceuticals products have huge production with extensive usage as well as frequently detected in surface water and ground water. How can the adsorbent be put into the underground water? How to control costs?

Author Response

Reviewer 2

  1. Please mark the main peaks of the various functional groups in Fig. 1C.

Our response: Peaks are now indicated in the Figure S8 of the revised Supplementary material.

  1. From the Fig. 2, the sample are not impurity. Why?

Our response: XRD as well as SEM-EDS measurements (the latter ones have been added at the revised version as supplementary material Figures S1, S2 and S3) do not seem to imply the presence of possible inorganic phase impurities in the studied sample. Some organic content may indeed be present, though, as reflected also by the high ratio of elemental C. If the referee refers to the kind of ‘inhomogeneous size’ of that sample, we already write in the manuscript that ‘Figure 2 depicts the presence of polygonal particles with rough surfaces and sizes ranging from a few hundreds of nanometers to a few micrometers’. But such different sizes do not necessarily imply the presence of varying compositions-phases throughout the sample.

  1. Please provide the recycle data.

Our response: Recycle data is now given in table S1 of the revised supplementary material.

  1. In the introduction, the authors have stated “On the other hand, conventional adsorbents in the likes of zeolites [11, 12] 45 and/or novel engineered materials such as metal organic frameworks (MOFs) [13-15]” The MOFs have been extensively studied by scientists and engineers as a fascinating feature the exploration of their use in numerous scientific domains, such as energy storages, environmental pollution, sensing platforms, and catalysis, photocatalysis, oxidation, hydrogenation due to their diverse active sites. Some more updated refs may be considered, such as Micropor. Mesopor. Mat, 341(2022) 112098; Mater. Today. Commum., 2022, 31,103514; and Dalton Trans., 2021, 50, 18016–18026; Inorganics, 10(2022) 202; Chem. Eng. J., 2022, 433, 133857.

Our response: We thank the reviewer for the suggestion. References have been included, accordingly.

  1. The particles sizes for different samples should be counted.

Our response: In fact, we already give some information in the manuscript regarding the sizes of the different samples. In particular, we mention that: a) ‘Figure 2 depicts the presence of polygonal particles with rough surfaces and sizes ranging from a few hundreds of nanometers to a few micrometers’, b) ‘Irregular-shaped with coarse surface and polygonal particles with sizes in the range from a few hundreds of nanometers up to a few micrometers are spotted in the SEM image of Figure 4’ and c) ‘Figure 6 illustrates the prevalence of polygonal particles with size in the macroscale.’ Here we need to note that the SEM is not like the TEM. So it is not easy to measure one-by-one the size of hundreds of different particles within the same sample. With the TEM this would be perhaps easier, because TEM images are two-dimensional projections of the particles and allow the use of many different magnifications, both lower but also very high ones. SEM imaging is a bit different and has some limitations in this context. Still, by measuring the SEM size of several different particles within the studied samples, we were able to write the above-mentioned phrases in the manuscript. In addition, we kindly remind to the reviewer that we already present the crystalline grain size of the different samples, derived through the XRD Scherrer’s method.

  1. What about the precise content of Co and carbon species in the materials?

Our response: We thank the reviewer for this suggestion, we are sorry for our negligence in this point. In the revised version, we have added in the complete SEM-EDS spectra (that includes, Co/Zn and C) for all composites, as supplementary material figures S1, S2 and S3.

  1. Since pharmaceuticals products have huge production with extensive usage as well as frequently detected in surface water and ground water. How can the adsorbent be put into the underground water? How to control costs?

Our response: Τhe query of the reviewer regarding the remediation of underground water is really interesting. There is still a lot of room in order to turn such approaches more cost-effective and more straightforward and in a larger scale from the technical point of view. However, it is not really our expertise to predict how these applications can pass to the real-world methods. Nevertheless, several magnetically driven systems are promising for instance encapsulated pesticides on magnetic nanocarries have been proposed for 2 in 1 action; the control release of the pesticide together with the easily removal of residual pesticide by a magnetic field, thus were significantly improved and their risks to the environment were reduced (Chemical Engineering Journal, 2017, 328, 320e330). Probably, at least for the current stage of development of the water remediation field, research and real-world methods should focus on the improvement of the use of adsorbents for the purification of surface and ground water.

Round 2

Reviewer 2 Report

accepted.